# STLnet: Signal Temporal Logic Enforced Multivariate Recurrent Neural Networks

**Meiyi Ma, Ji Gao, Lu Feng, John Stankovic**
University of Virginia
{meiyi,jg6yd,lu.feng,stankovic}@virginia.edu

## Abstract

Recurrent Neural Networks (RNNs) have made great achievements for sequential prediction tasks. In practice, the target sequence often follows certain model properties or patterns (e.g., reasonable ranges, consecutive changes, resource constraint, temporal correlations between multiple variables, existence, unusual cases, etc.). However, RNNs cannot guarantee their learned distributions satisfy these properties. It is even more challenging for the prediction of large-scale and complex Cyber-Physical Systems. Failure to produce outcomes that meet these properties will result in inaccurate and even meaningless results. In this paper, we develop a new temporal logic-based learning framework, STLnet, which guides the RNN learning process with auxiliary knowledge of model properties, and produces a more robust model for improved future predictions. Our framework can be applied to general sequential deep learning models, and trained in an end-to-end manner with back-propagation. We evaluate the performance of STLnet using large-scale real-world city data. The experimental results show STLnet not only improves the accuracy of predictions, but importantly also guarantees the satisfaction of model properties and increases the robustness of RNNs.

## 1 Introduction

Deep Neural Networks (DNNs), especially Recurrent Neural Networks (RNNs) have great achievements for sequential prediction tasks and are broadly applied to support the decision making of Cyber-Physical Systems (CPSs) [6, 15, 26]. Usually, in CPSs, RNNs are applied to predict the changing of system states or their environment. Systems take actions based on the prediction to guarantee the safety and performance of the system. For example, the power plant predicts the usages of energy in the next few days and decides how much energy to generate. An event service predicts the population and traffic for a big concert and allocates police and security resources. Training RNNs for complex CPSs (i.e., creating a prediction model) such as for Smart Cities is difficult [17]. The models are not always robust, often subject to anomalies, and subject to erroneous predictions, especially when the predictions are projected into the future (errors grow over time).

On the other hand, the target sequence often follows specific model properties or patterns, which should also be followed by the predicted sequence. For example, power plants have maximum and minimum limits of energy that can be generated per day, and the changing of air quality is relevant to the changing of traffic volume in the past hour. However, RNNs have no way to guarantee that their estimated distributions satisfy these model properties, especially for the properties with multiple variables and temporal features. Failure to follow these model properties can result in inaccurate and even meaningless results, e.g., predicted traffic volume exceeding the road capacity and inaccurate population estimation due to ignorance of big events happening a few hours ago.

**Challenges:** It is very challenging to enforce multivariate RNNs to follow temporal model properties in a sequence prediction task. The optimization mechanism of RNNs (i.e., back propagate the loss between estimated value and target value individually in a predicted sequence at each time

unit without comparing the temporal correlation of the two sequences, thus lack an integrated view about the sequential predictions) causes the challenge of the networks to follow the temporal model properties. In addition, unlike classification problems, it is more difficult to find an alternative approximate sequence that satisfies the property for knowledge distillation.

**Contributions:** In this paper, we create a new temporal logic-based learning framework, called STLnet, to guide the RNN learning process with auxiliary knowledge of model properties and to produce a more robust model that can then be used for improved future predictions. Unlike existing approaches, STLnet enforces the predicted multivariate sequence to follow its model properties by treating the sequence (i.e. a trace) as a whole. We first identify six key types of model properties and formalize them using Signal Temporal Logic (STL) [5]. Following the idea of knowledge distillation [8], the STLnet framework is built with a teacher network and a student network. In the teacher network, we create a STL trace generator to generate a trace that is closest to the trace predicted by the student network and satisfies the model properties simultaneously. We also create algorithms to efficiently generate satisfaction traces tailored to deep learning processes. We evaluate the performance of STLnet by applying it to an LSTM [9] network and a transformer network [23] for multivariate sequential prediction. The experimental results show that STLnet significantly increases the satisfaction of different types model properties (by about 4 times) and further improves the prediction accuracy (by about 18.5%).

To the best of our knowledge, our framework is the first work that integrates signal temporal logic with deep neural networks for sequential prediction. The strength of temporal logic offers a much stronger power to control a sequential system and a more flexible way to specify various types of properties. Different from previous literature, our method also creates a practical way to ensure the satisfaction of the logic rules. STLnet can be applied to general deep models to perform multivariate time series prediction and can be trained in an end-to-end manner. STLnet increases the robustness of the deep learning models.

## 2   Model Property Formalization using Signal Temporal Logic

In this paper, we refer to model properties as the inherent properties, rules, or patterns followed by the output sequences of target models or systems. These model properties are usually already known by the system or defined by the users before prediction, e.g., constraints by the physical world, or rules followed by the application domains (e.g., robotics). In practice, we can also mine the properties from the models' historical behaviors [12]. Actively learning these model properties helps build more robust deep neural networks.

**Specification Language – Signal Temporal Logic:** In order to enforce RNNs to learn the model properties, we first formalize properties using a machine-understandable specification language. For multivariate sequence prediction, capturing the relations between variables on the temporal domain is the most important task. Therefore, we apply STL [5] to formalize the model properties. STL is a very powerful formalism used to specify temporal properties of discrete and continuous signals. In this paper, we target the properties of the outputs of RNN models, which are discrete-time signals. The syntax of an STL formula $\varphi$ is usually defined as follows (see Appendix for formal definition),

$$\varphi ::= \mu \mid \neg\varphi \mid \varphi_1 \wedge \varphi_2 \mid \varphi_1 \vee \varphi_2 \mid \Diamond_{(a,b)}\varphi \mid \Box_{(a,b)}\varphi \mid \varphi \mathbf{U}_{(a,b)}\varphi.$$

We call $\mu$ a signal predicate, which is a formula in the form of $f(x) \geq 0$ with a signal variable $x \in \mathcal{X}$ and a function $f : \mathcal{X} \to \mathbb{R}$. The temporal operators $\Box$, $\Diamond$, and $\mathbf{U}$ denote "always", "eventually" and "until", respectively. The bounded interval $(a, b)$ denotes the time interval of temporal operators. $\mathbf{U}$ can be expressed by $\Box$ and $\Diamond$, thus we only consider $\Box$ and $\Diamond$ in this work.

**Model Properties and Formalization:** Systems from different application domains have varied types of model properties. Focusing on the CPSs, we identify several critical types (not necessarily a complete list) of model properties for the key applications below. We give specific examples of properties under each type in Table 1.

- *Reasonable Range:* One of the most fundamental model properties is that the value of the sequence should always be within a reasonable range constrained by the system or physical world, such as the road capacity of vehicles, normal ambient temperature, etc. It is not trivial for RNNs to learn

Table 1: Examples of Model Properties and Their STL Formulas

| Property Type | Example | STL formula |
|---|---|---|
| Reasonable Range | The traffic volume on a road can never exceed the road capacity. | $\Box_{[0,24]}(x_1 < \alpha_1) \wedge \cdots \wedge \Box_{[0,24]}(x_n < \alpha_n)$ |
| Consecutive Changes | The number of people in a shopping mall should not increase or decrease more than 1000 in 10 min if exits number is less than 5. | $y < 5 \rightarrow \Box_{[0,10]}(\Delta x < 1000)$ |
| Resource Constraint | The total energy distributed to all buildings should be less than $e$. | $\Box_{[0,24]}\mathsf{sum}(x_1, \ldots, x_n) < e$ |
| Variable and Temporal Correlation | For two consecutive intersections on a one-way direction road, if there are 10 cars passing intersection A, then there should be at least 10 cars passing intersection B within the next 5 minutes. | $(x_1 > 10 \rightarrow \Diamond_{[0,5]}(x_2 > 10)) \wedge \cdots \wedge (x_n > 10 \rightarrow \Diamond_{[0,5]}(x_{n+1} > 10))$ |
| Existence | There should be at least 1 patrol car around school every day. | $\Diamond_{[0,24]}x_1 \geq 1 \wedge \cdots \wedge \Diamond_{[0,24]}x_n \geq 1$ |
| Unusual Cases | If there is a concert on Friday, the number of people in the nearby shopping mall will increase at least 200 within 2 hours. | $x_{\mathsf{Event}} = \mathsf{True} \wedge x_{\mathsf{Day}} = \mathsf{Fri} \rightarrow \Diamond_{[0,2]}\Delta x > 200.$ |

reasonable ranges since they could vary by variable, conditionally relate to another variable, and dynamically change over time.

- *Consecutive Changes:* For most applications in CPSs and other domains, the consecutive changes of the target model over a fixed period follow specific properties, such as pollution levels or traffic volume levels from one time period to the next are bounded.

- *Resource Constraint:* The target models are often constrained by the resources, such as the available police resources to deal with an accident, or the maximum energy allocated by several locations. The resource constraints could also change over time in a real deployment, and are not necessarily the same as the training data. RNNs are highly likely to produce inaccurate or wrong outcomes without adapting the prediction results based on resource constraints.

- *Variable and Temporal Correlation:* There are correlations between different variables or locations over time, some of which are already known or easily discovered before training the learning model. These include the differences in air quality levels of adjacent locations, correlations between the air quality levels and traffic volume in the past hour, etc.

- *Existence:* Existence is a prevalent property in practice, but extremely difficult for RNNs to predict. It specifies the case that at least one of the values in the sequence (eventually) satisfies a specific property, e.g., traffic will be back to normal within 30 min after resolving an accident.

- *Unusual Cases:* The outputs of CPSs are highly affected by the environment and sensitive to uncertainties. For some unusual cases, there is a limited amount of data available in the training set. It is necessary to specify and teach the networks to learn the properties of these unusual cases (e.g., the influence of accidents or events on the population).

In Table 1, we present examples of these model properties and how to formalize them using STL. As we can see from the examples, most of the properties have temporal features over a given period, e.g., $[0, 2]$ indicates the next 2 hours from the checking point, $[0, 24]$ indicates the next 24 hours (as checking every hour for the whole day). These properties describe the essential features of the systems with complex temporal dependency among multiple variables. Traditional RNNs have no mechanism to check or learn them explicitly.

## 3 Problem Formulation

With the model properties specified, we formally define the logic enforced learning problem. Let $\omega = \{\omega^1, \omega^2, \ldots, \omega^m\}$ denotes the target sequences of data with $m$ variables over a finite discrete time domain $\mathbb{T}$ such that for the $k$th variable, $\omega^k[t] = x_t^k$ at any time $t \in \mathbb{T}$. Let $x_{[0,i]}^k$ be a prefix of sequence $\omega^k$ over the time domain $\{t_0, ..., t_i\} \subseteq \mathbb{T}$, and let $x_{[i+1,n]}^k$ be a suffix of sequence $\omega^k$ over the time domain $\{t_{i+1}, ..., t_n\} \subseteq \mathbb{T}$, where $n$ denotes the total time instances, thus we denote the target sequence as $\omega^k = x_{[0,i]}^k x_{[i+1,n]}^k$. We have a deep learning prediction model $f$ with parameter

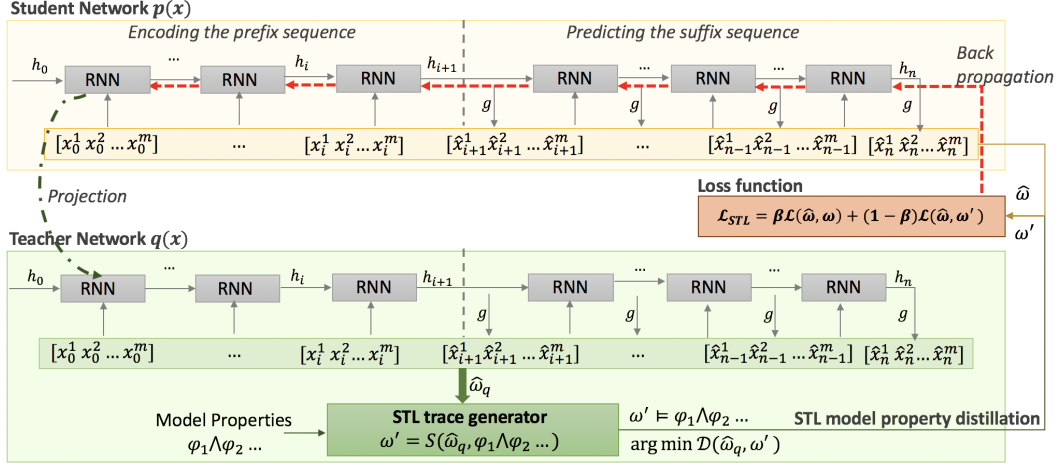

Figure 1: STLnet Framework

$\theta$ that predicts a suffix sequence with its prefix as inputs, i.e., $(\hat{x}^1_{[i+1,n]}, \hat{x}^2_{[i+1,n]}, \ldots, \hat{x}^m_{[i+1,n]}) = f((x^1_{[0,i]}, x^2_{[0,i]}, \ldots, x^m_{[0,i]}); \theta)$. We denote the predicted sequence as $\hat{\omega} = \{\hat{\omega}^1, \hat{\omega}^2, \ldots, \hat{\omega}^m\}$, where $\hat{\omega}^k = x^k_{[0,i]} \hat{x}^k_{[i+1,n]}$. Suppose the target sequence $\omega$ is drawn from a data distribution, i.e., $\omega \leftarrow \mathbf{w}$, and satisfies a set of properties, i.e., $\omega \models \varphi_1 \wedge \varphi_2 \wedge \ldots \wedge \varphi_\nu$, The goal is to find the model parameter $\theta$ that minimizes the distance (for a predefined distance metric $\mathcal{D}$) between the predicted sequence and the target sequence, and enforces the predicted sequence follows the same properties as well, i.e.,

$$\hat{\theta} = \arg\min_\theta \mathbb{E}_{\omega \leftarrow \mathbf{w}} \left[ \mathcal{D}(\omega, \hat{\omega}) \right]$$

$$s.t. \ \hat{\omega} \models \varphi_1 \wedge \varphi_2 \wedge \ldots \wedge \varphi_\nu$$

# 4 STLnet

Our solution is STLnet which enforces multivariate RNNs to return results that follow the model properties of the system. In this section, we first introduce the construction of STLnet in the training phase and show how to enforce the results to guarantee the satisfaction in the testing phase. Then, we present the STL trace generator, which is the key component of the teacher network.

## 4.1 STLnet Framework

Following the idea of knowledge distillation [8], the STLNet framework is built with a teacher network and a student network (as shown in Figure 1). The main idea is that whenever the student network fails to predict a trace (sequence) that follows the model properties, the teacher network generates a trace that is close to the trace returned by the student network and satisfies the model properties simultaneously. The student network then updates its parameters by learning from both the target trace and outcome of the teacher network.

In the *training* phase, our goal is to teach STLnet to learn from the "correct" traces, which include three major steps.

**Step 1 - Student network construction**: To start with, we build the basic student network, i.e., a general multivariate RNN $f$ (e.g., LSTM, GRU, Bi-LSTM, etc.). It takes the past states as inputs and predict their future states in $n$ time units, $(\hat{x}^1_{[i+1,n]}, \hat{x}^2_{[i+1,n]}, \ldots, \hat{x}^m_{[i+1,n]}) = f((x^1_{[0,i]}, x^2_{[0,i]}, \ldots, x^m_{[0,i]}); \theta)$. We denote the predicted sequence as $\hat{\omega} = \{\hat{\omega}^1, \hat{\omega}^2, \ldots, \hat{\omega}^m\}$, where $\hat{\omega}^k = x^k_{[0,i]} \hat{x}^k_{[i+1,n]}$ (i.e., the yellow box in Figure 1).

**Step 2 - Teacher network construction:** Next, we construct the teacher network $q(x)$ to generate a trace that satisfies the model properties $\varphi_1 \wedge \varphi_2 \wedge \cdots \wedge \varphi_\nu$ and has the shortest distance to the original

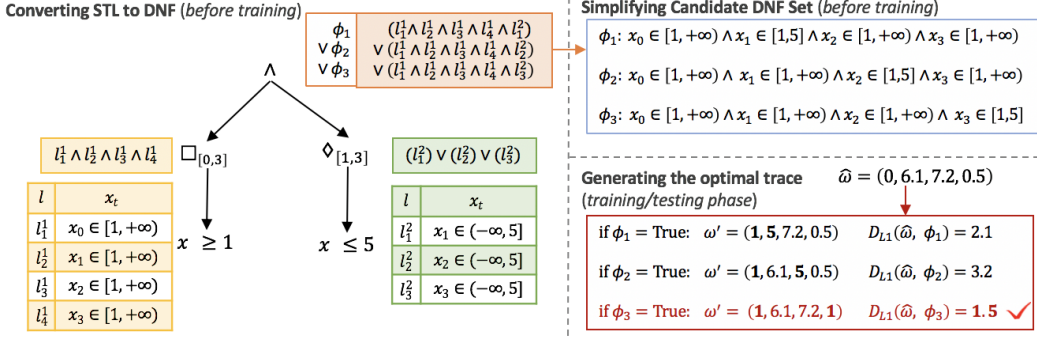

Figure 2: An example of STL trace generator ($\varphi = \square_{[0,3]} x \geq 1 \wedge \Diamond_{[1,3]} x \leq 5$)

prediction. We first formalize the model properties using STL. $q$ is constructed by projecting $p$ into a subspace constrained by the properties. Different than the structure of the student network $p$, $q$ has an STL trace generator. The STL trace generator first checks if this trace follows the properties, if yes, then output the trace $\omega' = \hat{\omega}$. If not, it generates a new trace $\omega'$, which first follows the properties, $\omega' \models \varphi_1 \wedge \varphi_2 \cdots \wedge \varphi_m$ and secondly, it is the closest trace to the original predicted trace $\hat{\omega}$. Here we use $L_1$ distance to measure the distance between two traces, i.e., the total amount of changes. We present the details of STL trace generator in Section 4.2.

**Step 3 - Back propagation with loss $\mathcal{L}_{\text{STL}}$:** The loss function is constructed by two parts to guide the student network $p(x)$ to balance between emulating the teacher's output and predicting the target trace. The target trace is $\omega$, thus the first part of the loss is $\mathcal{L}(\hat{\omega}, \omega)$, where $\mathcal{L}$ calculates the L-2 distance between two traces. The second part of the loss function is defined by the L-2 distance between the predicted trace $\hat{\omega}$ and teacher's output $\omega'$, i.e., $\mathcal{L}(\hat{\omega}, \omega')$. Thus, the student network is back propagated using the loss function as,

$$\mathcal{L}_{\text{STL}} = \beta \mathcal{L}(\hat{\omega}, \omega) + (1 - \beta)\mathcal{L}(\hat{\omega}, \omega') \tag{1}$$

The network is trained iteratively by repeating Steps 2 and 3 until convergence.

Similar to other distilled networks [10], in the *testing* phase, we can use either the distilled student network $p$ or the teacher network $q$ after a final projection. Our results show that both models substantially improve over the base network that is trained without STL specified properties. In practice, $q$ can guarantee the satisfaction of model properties while $p$ is more lightweight and efficient. We compare the performance of $p$ and $q$ with baseline network extensively in the evaluation.

## 4.2 STL Trace Generator

In this subsection, we introduce the algorithms of the key component of the teacher network, i.e., the STL trace generator. It is easy to check if a trace satisfies a specific property, however, to train the network, STLnet also needs to obtain the closest satisfying trace when the prediction results violate a property. It is a very difficult and time-consuming task.

In this paper, tailoring to the deep learning process, we create a STL trace generator. The key idea is to obtain a small Disjunctive Normal Form (DNF, a canonical normal form of a logical formula consisting of a disjunction of conjunctions) set representing the possible satisfaction ranges of each time on the trace before the optimization of the Deep Learning model. Then, we obtain the closest satisfaction trace of each instance in the training and testing phase. A simplified example of STL trace generator with a single variable is demonstrated in Figure 2.

**Converting STL to DNF** We first convert the STL formula into DNF and calculate the satisfaction range for each time unit on the sequence (left part of Figure 2). A nice property of a DNF representation is that for a trace to satisfy the requirement, it is a sufficient and necessary condition for it to match some clause $\phi$ inside $\varphi$. Therefore, we can check the properties in a straightforward manner by comparing the distance of the trace to each of the clauses in formula $\varphi$.

**Algorithm 1** Converting STL to DNF with Calculation of Satisfaction Range

---

1: **function** CalDNF($\varphi, t, $sgn)
2:   Input:STL Formula $\varphi$, time $t$, sign sgn
3:   Output: DNF Set $\xi$ representing the satisfaction range
4:   **if** sgn = False **then**
5:     **switch** $\varphi$ **do**
6:       **case** $\mu$
7:         **return** $\{x_t^j \mid f(x_t^j) < 0\}$;
8:       **case** $\neg\varphi$
9:         **return** CalDNF($\varphi, t, $True);
10:       **case** $\varphi_1 \wedge \varphi_2$
11:         **return** CalDNF($\neg\varphi_1 \vee \neg\varphi_2, t, $True);
12:       **case** $\varphi_1 \vee \varphi_2$
13:         **return** CalDNF($\neg\varphi_1 \wedge \neg\varphi_2, t, $True);
14:       **case** $\square_T\varphi$
15:         **return** CalDNF ($\lozenge_T\neg\varphi, t, $True);
16:       **case** $\lozenge_T\varphi$
17:         **return** CalDNF ($\square_T\neg\varphi, t, $True)
18:   **else**
19:     **switch** $\varphi$ **do**
20:       **case** $\mu$
21:         **return** $\{x_t^j \mid f(x_t^j) \geq 0\}$;
22:       **case** $\neg\varphi$
23:         **return** CalDNF($\varphi, t, $False)
24:       **case** $\varphi_1 \wedge \varphi_2$
25:         $\xi \leftarrow \emptyset$;
26:         **for** $\phi_1 \in$ CalDNF($\varphi_1, t, $sgn) **do**
27:           **for** $\phi_2 \in$ CalDNF($\varphi_2, t, $sgn)
      **do**
28:             $\xi \leftarrow \xi \vee (\phi_1 \wedge \phi_2)$;
29:           **end for**
30:         **end for**
31:         **return** $\xi$;
32:       **case** $\varphi_1 \vee \varphi_2$
33:         $\xi_1 \leftarrow$ CalDNF($\varphi_1, t, $sgn) ;
34:         $\xi_2 \leftarrow$ CalDNF($\varphi_2, t, $sgn) ;
35:         **return** $\xi_1 \vee \xi_2$;
36:       **case** $\square_T\varphi$
37:         $\xi \leftarrow \{$True$\}$
38:         **for** $t \in T$ **do**
39:           $\xi_1 \leftarrow$ CalDNF($\varphi_1, t, $sgn) ;
40:           $\xi_2 \leftarrow \emptyset$;
41:           **for** $\phi_1 \in \xi$ **do**
42:             **for** $\phi_2 \in \xi_1$ **do**
43:               $\xi_2 \leftarrow \xi_2 \vee (\phi_1 \wedge \phi_2)$;
44:             **end for**
45:           **end for**
46:           $\xi \leftarrow \xi_2$;
47:         **end for**
48:         **return** $\xi$;
49:       **case** $\lozenge_T\varphi$
50:         $\xi \leftarrow \emptyset$;
51:         **for** $t \in T$ **do**
52:           $\xi \leftarrow \xi \vee$ CalDNF($\varphi, t, $sgn);
53:         **end for**
54:         **return** $\xi$;
55:   **end if**
56: **end function**

---

**Proposition 4.1** (STL formula in DNF representation). *Every STL $\varphi$ can be represented in the DNF formula $\xi(\varphi)$, where $\xi(\varphi)$ is a formula that includes several clauses $\phi_k$ that are connected with the disjunction operator, and the length of $\phi_k$ is denoted by $|\phi_k|$. Each clause $\phi_k$ can be further represented by several Boolean variables $l_i$ that are connected with the conjunction operator. Finally, each Boolean variable $l_i$ is the satisfaction range of a specific parameter.*

$$
\begin{aligned}
\xi(\varphi) &= \phi_1 \vee \phi_2 \vee ... \vee \phi_K \\
\phi_k &= l_1^{(k)} \wedge l_2^{(k)} \wedge .. \wedge l_{|\phi_k|}^{(k)} \quad \forall k \in \{1, 2..K\} \\
l_i^{(k)} &= \{x_t^j \mid f(x_t^j) \geq 0\} \text{ where } (t \in T), \forall i \in \{1, 2..|\phi_k|\}
\end{aligned}
\tag{2}
$$

Algorithm 1 shows how to construct the DNF representation of the STL formula. The algorithm follows a top-down recursive manner. For every operator and its corresponding sub-tree, we first calculate the DNF formula of its children sub-trees, and then combine them with the operator. Specifically, the negation operator causes the DNF formula to become a CNF. Therefore, here we use De Morgan rule to sink the negation operator to the bottom level. Each clause of the DNF formula is guaranteed to have no duplicate variables. To be noted, if the set returned is an empty set, we know that the input requirement is unsatisfiable and we don't progress to the next steps. The computation time of Algorithm 1 is relevant to the number of predicates in the STL formula. We only execute it once in the pre-processing step before the training phase, thus it tailors to deep learning processes efficiently.

**Simplifying Candidate DNF Set:** In order to further obtain a smaller manageable DNF set, we reduce the size of the DNF set by finding the overlaps between the clauses. We first define the distance between a trace and a clause in DNF. (Note that we use $L1$ distance in the definition, which can be extended to any $Lp$ distance measure.)

**Definition 1** ($L1$-Distance of a trace to a clause). *Let clause $\phi = l_1 \wedge l_2 \wedge .. \wedge l_m$. The distance between a trace $\omega$ and clause $\phi$ is defined as*

$$D_{L1}(\omega, \phi) = \min_{\omega'} \sum_{t=1}^{T} |\omega'_t - \omega_t|, \tag{3}$$

$$where \ \ \omega' \models l_i \quad \forall i \in \{1, 2..m\}$$

The return value is a non-negative real number. If a variable satisfies a constrain in a clause, the term will be evaluated to 0; Otherwise, it will return the minimal distance over all the items in the satisfaction of $l_i$ (not necessary to be 1).

**Proposition 4.2** (The order of set). *For two clauses $\phi_i$ and $\phi_j$ in a DNF $\xi$, if $\forall \omega \models (\phi_i), \omega \models \phi_j$, and $\phi_i \subseteq \phi_j$, then we have $D_{L1}(\omega, \phi_i) \leq D_{L1}(\omega, \phi_j)$.*

If the satisfaction set of one clause is the subset of the satisfaction set of another clause, the first clause is unnecessary, as stated in Proposition 4.2. Therefore, we provide a pairwise comparison between all clauses, and we can remove some of the clauses and obtain a smaller set of DNF before training.

**Generating the optimal trace**    To satisfy a DNF representation, at least one of the clauses $\phi_k$ needs to be satisfied. Therefore, the best $\omega'$ is found by a specific $k$, as formally stated in Proposition 4.3.

**Proposition 4.3** (Shortest distance of a trace to the DNF formula). *Let $\hat{\omega}$ be the trace that satisfy the DNF formula $\varphi = \phi_1 \vee \phi_2 \vee ... \vee \phi_K$ that has minimal distance to the input trace $\omega$, then we have*

$$\hat{k} = \arg \min_k D_{L1}(\omega, \phi_k) \tag{4}$$

*and $\hat{\omega}$ is the trace that minimizes $D_{L1}(\omega, \phi_{\hat{k}})$ by $D_{L1}(\omega, \hat{\omega}) = D_{L1}(\omega, \phi_{\hat{k}})$.*

For each clause in the DNF set, we calculate the distance between the trace to be optimized with the clause. The distance can be then calculated by a summation over the distance of the satisfaction of all the Boolean variables. After the distance calculation, we return the optimal trace with the minimal distance as our generated target trace. The returned trace is therefore guaranteed to satisfy the requirement. In the example given in Figure 2 (right side), if the trace predicted by the student network is $\hat{\omega} = (0, 6.1, 7.2, 0.5)$, then, the optimal new trace generated by the teacher network is $\omega' = (\mathbf{1}, 6.1, 7.2, \mathbf{1})$, which has the shortest distance to $\hat{\omega}$ and satisfies its model property.

## 5   Evaluation

We evaluate the performance of STLnet from two aspects: the capability of learning different types of model properties and the performance in learning from a city dataset with multiple variables.

In all experiments, we evaluate the performance using three *metrics*, i.e., Root Mean Square Error (RMSE), property satisfaction rate, and average STL robustness value $\rho$. RMSE measures the accuracy of the prediction, property satisfaction rate shows the percentage of the predicted sequence that satisfies the property, and STL robustness value $\rho$ shows the degree of satisfaction (we refer paper [5] or our supplementary materials for the definition of quantitative semantics). Briefly, if $\rho \leq 0$, the property is violated. The smaller $\rho$ is, the more the property is violated.

To evaluate the performance of STLnet, we applied it to two networks, i.e., an LSTM network [9] and a transformer network [23] for multivariate sequential prediction. For each model, we compare the experimental results among three networks, i.e., general model, model with STLnet testing with the student network ($p$), and model with STLnet testing with the teacher network ($q$). Applying STLnet-$q$ can guarantee the satisfaction of all model properties, so we also present the results of STLnet-$p$ here to show the improvement we achieved through training. To be noted, STLnet is general and can apply to all RNNs. We use LSTM and Transformer networks as examples. The experiments are evaluated on a server machine with 20 CPUs, each core is 2.2GHz, and 4 Nvidia GeForce RTX 2080Ti GPUs. The operating system is Centos 7.

Table 2: Comparison of Accuracy and Property Satisfaction among LSTM, STLnet-$p$ and STLnet-$q$

| | LSTM | | | LSTM STLnet-$p$ | | | LSTM STLnet-$q$ | | |
|---|---|---|---|---|---|---|---|---|---|
| | RMSE | Sat Rate | Violate$\rho$ | RMSE | Sat Rate | Violate$\rho$ | RMSE | Sat Rate | Violate$\rho$ |
| $\varphi_1$ | 0.026 | 92.00% | -0.298 | 0.025 | 98.34% | -0.014 | 0.025 | 100.00% | 0 |
| $\varphi_2$ | 94.304 | 75.61% | -117.982 | 90.016 | 97.78% | -1.603 | 90.160 | 100.00% | 0 |
| $\varphi_3$ | 4.214 | 75.47% | -1.589 | 4.209 | 87.69% | -0.606 | 4.209 | 100.00% | 0 |
| $\varphi_4$ | 0.309 | 56.68% | -36.884 | 0.230 | 83.09% | -3.906 | 0.229 | 100.00% | 0 |
| $\varphi_5$ | 2.188 | 0.84% | -463.534 | 1.151 | 75.64% | -19.842 | 1.162 | 100.00% | 0 |
| $\varphi_6$ | 8.603 | 59.54% | -282.403 | 8.532 | 61.85% | -282.403 | 7.122 | 100.00% | 0 |

Table 3: Comparison of Accuracy and Property Satisfaction among Transformer Model, STLnet-$p$ and STLnet-$q$

| | Transformer | | | Transformer STLnet-$p$ | | | Transformer STLnet-$q$ | | |
|---|---|---|---|---|---|---|---|---|---|
| | RMSE | Sat Rate | Violate$\rho$ | RMSE | Sat Rate | Violate$\rho$ | RMSE | Sat Rate | Violate$\rho$ |
| $\varphi_1$ | 0.045 | 27.76% | -18.808 | 0.031 | 89.48% | -1.835 | 0.031 | 100.00% | 0 |
| $\varphi_2$ | 105.211 | 49.44% | -109.282 | 111.688 | 76.08% | -18.874 | 111.655 | 100.00% | 0 |
| $\varphi_3$ | 4.340 | 52.96% | -3.855 | 4.339 | 60.70% | -2.596 | 4.339 | 100.00% | 0 |
| $\varphi_4$ | 0.124 | 0.36% | -38.893 | 0.135 | 51.00% | -5.101 | 0.135 | 100.00% | 0 |
| $\varphi_5$ | 2.196 | 8.88% | -31.172 | 1.805 | 50.20% | -4.612 | 1.804 | 100.00% | 0 |
| $\varphi_6$ | 8.156 | 20.08% | -301.175 | 8.326 | 20.32% | -307.165 | 2.657 | 100.00% | 0 |

## 5.1 Learning Model Properties

The *goal* of the first set of experiments is to show that STLnet is general and robust to different types of model properties and improves the satisfaction rate significantly. To start with, we synthesize six sets of data that satisfy six types of model properties, respectively. Due to the page limit, the detailed description of the synthesized data and model properties, and the datasets are provided in the supplementary materials.

The results shown in Table 2 and Table 3 are obtained from 25 runs. From the results, we can see that: (1) STLnet significantly improves the model property satisfaction rate for both LSTM and Transformer. For all the property, both satisfaction rate and violation degree are improved by STLnet. For example, property $\varphi_5$ and $\varphi_4$, the satisfaction rates of basic LSTM are only 0.84% and 56.68% with very high violation degrees (-463.534 and -36.884), while STLnet-$p$ achieves 75.64% and 83.09% satisfaction rate with violation degree dropping to -19.842 and -3.906, respectively. STLnet-$q$ can guarantee the satisfaction of all the model properties. (2) For $\varphi_1$ to $\varphi_6$, STNnet not only improves the satisfaction rate, but also decreases the RMSE value. In general, STL-q can further improve the accuracy. For example, property $\varphi_6$ (property type of unusual cases), RNNs are not able to learn the unusual cases which have a small portion of instances in the training data and lead to a low satisfaction rate. While STLnet guides the predicted trace follow the property (in both training and testing), it also decreases RMSE. It indicates that learning model properties can support RNNs to build a more accurate model. Overall, the results prove the effectiveness and generalizability of STLnet dealing with different types of model properties.

## 5.2 Multivariate Air Quality Prediction with Model Properties

The *goal* of the second set of experiments is to show how STLnet improves the accuracy and robustness of RNNs in a real-world CPS application, especially in cases of noisy/missing sensing data, and long term prediction. We apply STLnet to train RNN models with air quality datasets. The dataset includes 1.3 million instances of 6 pollutants (i.e., PM2.5, PM10, CO, $SO_2$, $NO_2$, $O_3$) collected from 130 locations in Beijing every hour between 5/1/2014 and 4/30/2015 [15]. To build the LSTM network, we regard one pollutant from one location as one variable, and concatenate all variables from the same time unit. Next, we specify important model properties, including reasonable ranges, consecutive changes, correlations between different pollutants, and between different locations, etc.

The results of the comparison are presented in Figure 3. From the results, we can see that, (1) STLnet improves both property satisfaction rate (i.e., from 20% to over 70% on average) and RMSE (i.e., from about 150 to 130 on average). STLnet-$q$ outperforms STLnet-$p$ regarding the satisfaction rate

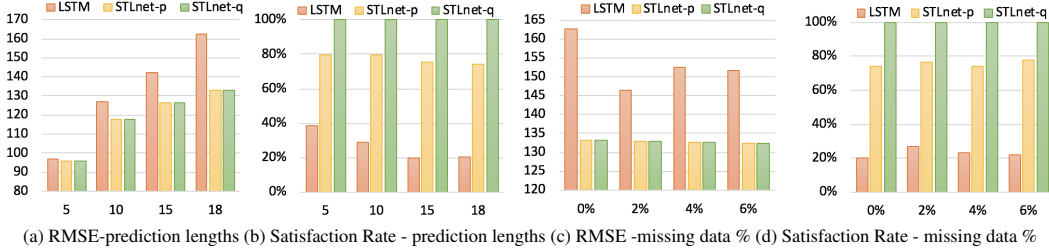

(a) RMSE-prediction lengths (b) Satisfaction Rate - prediction lengths (c) RMSE -missing data % (d) Satisfaction Rate - missing data %

Figure 3: Comparison of RMSE and Satisfaction Rate among LSTM, STLnet-$p$ and STLnet-$q$

and achieves a similar RMSE as STLnet-$p$. (2) Figure 3 (a) and (b) compare the performance with different time lengths of prediction. When predicting future 5-time units, three networks have a very similar RMSE value. With the prediction length increasing, the RMSE value of LSTM increases greatly. However, with STLnet, the prediction accuracy is improved, e.g., when $l = 18$, RMSE drops from 162 to 132 (18.5%). (3) Datasets with missing data affect the learning performance. We test the model performance with different percentages of missing data. The results show that STLnet is able to improve model accuracy by as much as 14% and property satisfaction rate by 3 to 4 times ($p$ and $q$, respectively). Overall, the results indicate the effectiveness and robustness of STLnet in a real-world application. It also can support RNN models to perform long-term prediction with missing data.

## 6   Related Work

The attempts of enforcing machine learning to follow logic rules are dated to the early stage of the development of the neural network. So-called Neural Symbolic Systems [22, 7] construct network architectures to combine inference with logic rules. Combining logic rules with various machine learning models has been successful [13, 27]. Breaking the black box of the neural network has always been a popular research topic. Applying logic rules, as one typical approach to break the black box, has attracted much attention. A direct solution is to formulate the logic rule as an optimizable loss item. By minimizing the logic loss, soft constraints of the logic rule are proposed to the model [25, 21, 20]. Other methods include Logistic circuits [14] and Logic Tensor Network [4] design specific structural to incorporate logic rules. [24] generates a graph model to embed logic rule into the prediction. Following knowledge distillation [8], [10] proposes a way to integrate rules defined by first-order logic with knowledge distillation. Following this method, several works [11, 3] propose ways to better train deep NLP models with some specific type of logic rules, which uses posterior regularization to constraint the student network. However, most previous works only apply simple and straightforward logic rules, target single variable classification problems (e.g., sentiment classification), and only apply a soft constraint to (rather than a guarantee) the satisfaction. Different from previous work, our work targets multivariate sequential prediction models and guides them to learn the model properties with complex temporal features in regression tasks. Therefore, we chose *Signal Temporal Logic*, a logic variant that focuses on the temporal properties, to formalize the model properties. As a powerful specification language, STL has been broadly applied to the model checking, specification and verification for CPS applications [1], such as robotics [19], smart cities [16, 18], healthcare [2]. The introduction of temporal operators (e.g., always, eventually and until) makes STL more natural, intuitive and flexible in describing dynamic systems. STL has been applied to both continuous and discrete signals.

## 7   Conclusion

In order to guide Multivariate RNNs to follow the model properties of the system and produce a more robust model for improved future predictions, we build STLnet, a new temporal logic-based learning framework. The experimental results show that STLnet not only improves the accuracy of predictions, but importantly also guarantees the satisfaction of model properties. The promising results also indicate that considering model properties is very important for building deep learning models for complex systems, and formal logic can be an effective way to enhance the robustness of the deep learning models.

## Broader Impact

The approach created in this paper can be broadly applied to the tasks of sequence prediction using RNN models in application domains such as smart cities, smart health, and other CPS-IoT systems. In these systems prediction results are usually used to support monitoring and decision making processes. The goal is to improve the prediction accuracy and, more importantly, guarantee the satisfaction of critical properties. We envision that relevant systems and decision-makers will benefit from this work. In this way smart cities and smart health systems can improve safety and performance, thereby improving daily life and health for people. Failure of the system (i.e., the model produces wrong prediction results) could affect the decisions made based on the results. In practice, even if the prediction results are somewhat inaccurate (e.g., high RMSE), STLnet can still guarantee the satisfaction of key properties.

## Acknowledgments and Disclosure of Funding

This research was partially supported by NSF grants CCF-1942836 and CNS-1739333, and the Commonwealth Cyber Initiative, an investment from the Commonwealth of Virginia in the advancement of cyber R&D, innovation, and workforce development.

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
