[Supplementary Material]



# Supplementary Materials

## 1 Overview

In the supplementary materials for paper "STLnet: Signal Temporal Logic Enforced Multivariate Recurrent Neural Networks", we first introduce the preliminaries on Signal Temporal Logic in Section 2; Next, we give the formal proofs for the three propositions we proposed in the paper (Section 3); Finally, we present more details of evaluation (Section 4). We also include the code of STLnet and the synthesized datasets in the .zip file.

## 2 Preliminaries: Signal Temporal Logic

To briefly introduce the syntax and semantics of STL, we denote by $X$ and $P$ finite sets of real and propositional variables. We let $\omega : \mathbb{T} \to \mathbb{R}^m \times \mathbb{B}^n$ be a multi-dimensional signal, where $\mathbb{T} = [0, d) \subseteq \mathbb{R}$, $m = |X|$, $n = |P|$. Given a variable $v \in X \cup P$, we denote by $\pi_v(\omega)$ the projection of $\omega$ on its component $v$. The syntax of an STL formula $\varphi$ is usually defined as follows,

$$\varphi ::= \mu \mid \neg\varphi \mid \varphi \wedge \varphi \mid \Diamond_{(a,b)}\varphi \mid \Box_{(a,b)}\varphi \mid \varphi \mathbf{U}_{(a,b)}\varphi.$$

We call $\mu$ a signal predicate, which is a formula in the form of $f(x) > 0$ with a signal variable $x \in \mathcal{X}$ and a function $f : \mathcal{X} \to \mathbb{R}$. The temporal operators $\Box$, $\Diamond$, and $\mathbf{U}$ denote "always", "eventually" and "until", respectively. The bounded interval $(a, b)$ denotes the time interval of temporal operators.

Below we present the formal definition of STL Boolean semantics. To informally explain the STL operations, formula $\Box_{(a,b)}\varphi$ is true iff $\varphi$ is always true in the time interval $(a, b)$. Formula $\Diamond_{(a,b)}\varphi$ is true iff $\varphi$ is true at sometime between $a$ and $b$. Formula $\varphi_1 \mathbf{U}_{(a,b)}\varphi_2$ is true iff $\varphi_1$ is true until $\varphi_2$ becomes true at sometime between $a$ and $b$.

$$
\begin{aligned}
(\omega, t) &\models \mu & \Leftrightarrow & \quad f(x) > 0 \\
(\omega, t) &\models \neg\varphi & \Leftrightarrow & \quad (\omega, t) \models \varphi \\
(\omega, t) &\models \varphi_1 \wedge \varphi_2 & \Leftrightarrow & \quad (\omega, t) \models \varphi_1 \text{ and } (\omega, t) \models \varphi_2 \\
(\omega, t) &\models \Box_{(a,b)} & \Leftrightarrow & \quad \forall t \in (a, b), (\omega, t) \models \varphi \\
(\omega, t) &\models \Diamond_{(a,b)} & \Leftrightarrow & \quad \exists t \in (a, b) \cap \mathbb{T}, (\omega, t) \models \varphi \\
(\omega, t) &\models \varphi_1 \mathbf{U}_I \varphi_2 & \Leftrightarrow & \quad \exists t' \in (t + a, t + b) \cap \mathbb{T}, (\omega, t') \models \varphi_2 \text{ and } \forall t'' \in (t, t'), (\omega, t'') \models \varphi_1
\end{aligned}
$$

Next, we present the formal definition of STL quantitative semantics.

$$
\begin{aligned}
\rho(x \sim c, \omega, t) &= \pi_x(\omega)[t] - c \\
\rho(\neg\varphi, \omega, t) &= -\rho(\varphi, \omega, t) \\
\rho(\varphi_1 \wedge \varphi_2, \omega, t) &= \min\{\rho(\varphi_1, \omega, t), \rho(\varphi_2, \omega, t)\} \\
\rho(\Box_I \varphi, \omega, t) &= \min_{t' \in (t, t+I)} \rho(\varphi, \omega, t') \\
\rho(\Diamond_I \varphi, \omega, t) &= \max_{t' \in (t, t+I)} \rho(\varphi, \omega, t') \\
\rho(\varphi_1 \mathbf{U}_I \varphi_2, \omega, t) &= \sup_{t' \in (t+I) \cap \mathbb{T}} (\min\{\rho(\varphi_2, \omega, t'), \inf_{t'' \in [t, t']} (\rho(\varphi_1, \omega, t''))\})
\end{aligned}
$$

The quantitative semantics (i.e., the robustness values) measure the satisfaction/violation degree of the STL formula. In the evaluation section of the paper, we use it to measure the prediction performance on property satisfaction.

## 3 Proof of Propositions

**Proposition 4.1** (Restate, STL formula in DNF representation). *Every STL $\varphi$ can be represented in the DNF formula $\xi(\varphi)$, where $\xi(\varphi)$ is a formula that includes several clauses $\phi_k$ that is connected with the disjunction operator, and the length of $\phi_k$ is denoted by $|\phi_k|$. Each clause $\phi_k$ can be further represented by several Boolean variables $l_i$ that are connected with the conjunction operator. Finally, each Boolean variable $l_i$ is the satisfaction range of a specific parameter.*

$$
\begin{aligned}
\xi(\varphi) &= \phi_1 \vee \phi_2 \vee ... \vee \phi_K \\
\phi_k &= l_1^{(k)} \wedge l_2^{(k)} \wedge .. \wedge l_{|\phi_k|}^{(k)} \quad \forall k \in \{1, 2..K\} \\
l_i^{(k)} &= \{x_t^j \mid f(x_t^j) \geq 0\} \text{ where } (t \in T), \forall i \in \{1, 2..|\phi_k|\}
\end{aligned}
\tag{1}
$$

*Proof.* We prove Proposition 4.1 by induction. We use induction on the top-layer operator:

- A single $\mu$ operator can be represented by a single $l$ clause, where $f(x_0) \geq 0$.

- If the low layer operators can be represented by DNF formula, the result of $\neg$, $\wedge$, and $\vee$ operators can also be represented as DNF formula by the De Morgen rule.

- The always operator $\square_{(a,b)}\phi$ can be decomposed as multiple $\wedge$ operator on the time period $(a,b)$. Given the DNF $\varphi$ and a specific time $t \in (a,b)$, the actual DNF should be $\varphi$ with an additive time shift $t$ on every time operator in $\varphi$. Then the STL formula is equivalent to a DNF built by applying the De Morgen rule on the DNFs with every $t \in (a,b)$.

- The eventually operator $\lozenge_{(a,b)}\phi$ can be decomposed as multiple $\vee$ operator on the time period $(a,b)$. Given the DNF $\varphi$ and a specific time $t \in (a,b)$, the actual DNF should be $\varphi$ with an additive time shift $t$ on every time operator in $\varphi$. Then the STL formula is equivalent to a DNF built by connecting the DNFs for every $t \in (a,b)$ with $\vee$.

- The until operator $\mathbf{U}_{(a,b)}$ by the STL definition can be represented with $\square$ and $\lozenge$ operators. Therefore it can also be represented by a DNF.

By induction, we have Proposition 4.1 proved. $\square$

**Proposition 4.2** (Restate). *For two clauses $\phi_i$ and $\phi_j$ in a DNF $\xi$, if $\forall \omega \models \phi_i, \omega \models \phi_j$, and $\phi_i \subseteq \phi_j$, then we have $D_{L1}(\omega, \phi_i) \leq D_{L1}(\omega, \phi_j)$.*

*Proof.* Prove by contradiction. Assume $D_{L1}(\omega, \phi_i) > D_{L1}(\omega, \phi_j)$. Let $\omega'$ denotes the trace with minimal distance to $\omega$ in $\phi_j$, that is, $\omega' = \arg\min_{\omega'' \models \phi_j} D_{L1}(\omega, \omega'')$. As $\phi_i \subseteq \phi_j$, we have $\omega' \models \phi_i$. Therefore, $D_{L1}(\omega, \phi_i) = \min_{\omega'' \models \phi_i} D_{L1}(\omega, \omega'') \leq D_{L1}(\omega, \omega')$, which clearly contradicts the assumption. Therefore, $D_{L1}(\omega, \phi_i) \leq D_{L1}(\omega, \phi_j)$. $\square$

**Proposition 4.3** (Restate, shortest distance of a trace to the DNF formula). *Let $\hat{\omega}$ be the trace that satisfy the DNF formula $\varphi = \phi_1 \vee \phi_2 \vee ... \vee \phi_K$ that has minimal distance to the input trace $\omega$, then we have*

$$\hat{k} = \arg\min_k D_{L1}(\omega, \phi_k) \tag{2}$$

*and $\hat{\omega}$ is the trace that minimizes $D_{L1}(\omega, \phi_{\hat{k}})$ by $D_{L1}(\omega, \hat{\omega}) = D_{L1}(\omega, \phi_{\hat{k}})$.*

*Proof.* We prove the proposition by contradiction. Assume $\hat{\omega} \models \varphi$, by the definition of DNF formula, we have $\exists k : \hat{\omega} \models \phi_k$. Suppose $\hat{k}$ is one of choices that $\hat{\omega} \models \phi_{\hat{k}}$.

If $\hat{k} \neq \arg\min_k D_{L1}(\omega, \phi_k)$, then there exists another $k'$ that $D_{L1}(\omega, \phi_{k'}) < D_{L1}(\omega, \phi_{\hat{k}})$. By the definition of $D_{L1}(\omega, \phi_{k'})$, there exists a $\omega'$ that $D_{L1}(\omega, \omega') = D_{L1}(\omega, \phi_{k'}) < D_{L1}(\omega, \phi_{\hat{k}}) = D_{L1}(\omega, \hat{\omega})$. We also have $\omega' \models \phi_{k'}$, which indicates $\omega' \models \varphi$. Then $\omega'$ is closer to $\omega$ and also satisfies $\varphi$, which contradicts the assumption.

If $\hat{\omega}$ doesn't minimize $D_{L1}(\omega, \phi_{\hat{k}})$, then there exists another $\omega' \models \phi_{\hat{k}}$ that $D_{L1}(\omega, \omega') = D_{L1}(\omega, \phi_{\hat{k}}) < D_{L1}(\omega, \hat{\omega})$. Then $\omega'$ is closer to $\omega$ and also satisfies $\varphi$, which contradicts the assumption. $\square$

# 4  Evaluation

In Section 5.1 of the paper, we present the results of the learning model properties from six sets of synthesized datasets to show how STLnet support RNNs to better learn model properties. Here we elaborate the details of how we synthesized the datasets and their model properties. As we can see, all the six synthesized datasets are abstracted from very common scenarios from CPSs applications.

For each of the six sets of experiments, we generated 50,000 instances ($n_d$) and divided them into five subsets. Then, for each subset, we randomly selected 95% for training and 5% for testing. We repeated it five times. At last, we calculated the average results from these 25 runs.

Below we present STL formulas of the model properties for each set of datasets. We also explained how we synthesized the datasets.

- *Resource constraint:*

  To synthesize the data with the model property of resource constraint, we use a piecewise constant function to generate $n_d$ instances, each following:

  $$x_1(t) = x_2(t) = \begin{cases} 1.0 - \sigma(t) & t < d \\ 1.005 + \sigma(t) & t \geq d. \end{cases} \tag{3}$$

  where $\sigma(t)$ is a small Gaussian noise, and $d$ is pick randomly between 10 to 14. The function follows model property $\varphi_1$, which is used in STLnet to enhance learning,

  $$\varphi_1 = \Box_{[0,8]} \neg(x_1 > 1) \wedge \Box_{[14,19]}(x_1 > 1) \wedge \Box_{[0,8]} \neg(x_2 > 1) \wedge \Box_{[14,19]}(x_2 > 1). \tag{4}$$

- *Consecutive change:*

  To synthesize the data with the model property of consecutive change, we use a monotonically decreasing function to generate $n_d$ sequences, each following:

  $$\begin{aligned} x_1(t) &= x_1(t-1) - \min(100, 0.2x_1(t-1)) \\ x_2(t) &= x_2(t-1) - \min(100, 0.2x_2(t-1)). \end{aligned} \tag{5}$$

  We pick the original value $x_1(0)$ and $x_2(0)$ uniformly between the range [0, 1000]. The function follows model property $\varphi_2$, which is used in STLnet to enhance learning,

  $$\varphi_2 = \Box_{[0,19]}(\neg(\Delta x_1 > 100) \wedge \neg(\Delta x_2 > 100)). \tag{6}$$

- *Variable and Temporal Correlation:*

  To synthesize the data with the model property of variable and temporal correlation, we generate to generate $n_d$ sequences. Each sequence consists only 0 and 1, but keep not any group of 4 consecutive numbers to be the same. That is,

  $$x_1(t) = \begin{cases} 0 & \text{If } x_1(t-1) = 1 \wedge x_1(t-2) = 1 \wedge x_1(t-3) = 1 \\ 1 & \text{If } x_1(t-1) = 0 \wedge x_1(t-2) = 0 \wedge x_1(t-3) = 0 \\ \text{Bernoulli(0.5)} & \text{Otherwise.} \end{cases} \tag{7}$$

  The function follows model property $\varphi_3$, which is used in STLnet to enhance learning,

  $$\varphi_3 = \Box_{[0,5]} \left( \Diamond_{[0,4]}(x_1 > 0) \wedge \Diamond_{[0,4]}(\neg(x_1 > 0)) \right). \tag{8}$$

- *Reasonable range:*

  To synthesize the data with the model property of reasonable range, we use a periodic function to generate $n_d$ sequences, each following:

  $$\begin{aligned} x_1(t) &= \sin(at + b) \\ x_2(t) &= \cos(at + b). \end{aligned} \tag{9}$$

  Where $a$ is uniformly picked from $[0.77, 1.03)$, and $b$ is uniformly picked from $[0, 0.5)$. The function follows model property $\varphi_4$, which is used in STLnet to enhance learning,

  $$\varphi_4 = \Box_{[0,19]}(x_1 > -1.0 \wedge \neg(x_1 > 1.0) \wedge x_2 > -1.0 \wedge \neg(x_2 > 1.0)). \tag{10}$$

- *Existence:*

  To synthesize the data, we generate $n_d$ instances of 0 and 1. In each sequence make sure that for both $x_1$ and $x_2$ it equals 1 at a single $t$ and equals 0 at other time. The function follows model property $\varphi_5$, which is used in STLnet to enhance learning,

  $$\varphi_5 = \Diamond[0, 19](x_1 > 0.99) \wedge \Diamond[0, 19](x_2 > 0.99). \tag{11}$$

- *Unusual cases:*

  To synthesize the data with the model property of unusual cases, we generate $n_d$ instances following:

  $$x_1(t) = \begin{cases} 1000 & t = t_d \\ 0 & \text{otherwise.} \end{cases} \tag{12}$$

  and

  $$x_2(t) = \begin{cases} 10 & \exists t_i \in [1, 9], x_1(t - t_i) > 0 \\ \sigma(t) & \text{otherwise.} \end{cases} \tag{13}$$

  where $d$ is pick randomly between 0 to 4, and $\sigma(t)$ is a small Gaussian noise.

  The function follows model property $\varphi_6$, which is used in STLnet to enhance learning,

  $$\varphi_6 = \Box_{[0,4]}(x_1 > 500 \vee \Box_{[1,9]} x_2 > 9). \tag{14}$$