[Reviews · NeurIPS 2020]

Review 1

Summary and Contributions: The paper proposes a new approach of enforcing properties with the predictions RNN makes on sequence of data. The key idea builds upon knowledge distillation, in which a student network learns to predict a sequence that maximizes the posterior probability while teacher network generates a formally correct trace (w.r.t the properties to be enforced) that is the closest to the predicted trace to rein in the student network if its predicted trace violates the given properties.

Strengths: Fundamentally, it's a relatively straight-forward idea but it's nicely executed and very well presented. The evaluation points to the effectiveness of the proposed approach.

Weaknesses: I have several questions for the authors. (1) in Figure 1. it seems like the teacher network also employs a sequence network but if I understood the idea correctly, all teacher network does is to find a correct trace within constraints in the form of DNF (converted from the stl) that is the closet to the trace predicted by the student network. Could you please clarify? (2) I am a little confused about the distance metric in equation 3, specifically, the absolute term, does this mean if a variable satisfies a constrain in a clause, the term will be evaluated to 0 otherwise 1? (3) I am a little suspicious about the scalability of algorithm 1. By that I mean what happens if there is a large number of predicates in the STL as well as variables to be predicted in the sequence. Since the authors did not touch upon this aspect in the evaluation, it would be great if they can have a discussion on this matter. (4) Finally, a trivial question, why don't you work with transformer-based sequence models rather than RNN, LSTM given the dominance of the former?

Correctness: There is nothing I can find in the paper that pose a threat to the correctness of their approach.

Clarity: The paper is very well-written and easy to follow.

Relation to Prior Work: I think authors have done a good job to articulate their contributions w.r.t the prior arts.

Reproducibility: Yes

Additional Feedback: Did not check the supplemental material as I think the paper is over the bar. ----------Updates After Author Response---------- This paper is somewhat heavily contested. To some agree, I do appreciate the other reviewers' concern on the necessity of using STL instead of FOL. But I still believe the paper has merits and worth publication. For the final version, I would strongly recommend authors to add a more detailed explanation on your choice of using STL instead of FOL.


Review 2

Summary and Contributions: The authors propose a new learning architecture called STLnet. They use Signal Temporal Logic (STL) to ensure RNN predictions satisfy certain properties of Cyber Physical Systems. Important types of model properties are identified and formalised using STL. A teacher network generates a trace closest to the trace predicted by the student network satisfying the model properties. The experiments show STLnet using LSTMs is much better at satisfying the model properties than regular LSTMs.

Strengths: The propose architecture is innovative, and the way in which the authors combine temporal logic with deep learning is interesting. I agree with the general claim that it is challenging to ensure the output of ML models satisfies certain properties, and the way in which the authors propose to do this is quite convincing to me. Temporal logic seems like an appropriate way of specifying this, and the empirical results look promising.

Weaknesses: The paper contains a number of typos and mistakes that should be corrected, and the positioning in related work could be improved. === After rebuttal === I am quite satisfied with the replies of the authors, however during the reviewer discussions I did start to doubt the novelty of the approach. The system properties do not seem novel (they appear in the literature already), and the empirical results only compare with a regular LSTM. How does this relate to other comparable approaches?

Correctness: Yes.

Clarity: Yes, but there are a number of typos.

Relation to Prior Work: I am not familiar enough with the topic to judge this in detail, but I do think the authors' claim "To the best of our knowledge, our framework is the first work that integrates temporal logic with deep neural networks." is a bit strong. Searching for "temporal logic deep learning" on google scholar seems to contradict this claim, so they could position it in related work better.

Reproducibility: Yes

Additional Feedback: General comments: - 50: I didn't really get the challenge here. The optimization mechanism of autoregressive RNNs does take into account the predicted value (or target value, if teacher forcing is applied) at the previous time step. So I don't understand what you mean by "without comparing the temporal correlation of the two sequences as a whole". - 87: Here you say f(x) >= 0 but in the appendix f(x) > 0. - 91: Where do these properties come from? Did you invent them yourself? How did you get to them? - Table 1: reasonable range: I think you are missing a closing parenthesis for "(x_1 < \alpha_1)". Also the new-line is at a strange place. Also, do 0-24 represent hours in a day? This is not explained. - Table 1: resource constraint. Why are you using [0, 24] here, but (0, 24) in "reasonable range"? - Algorithm 1: the loop on line 38 seems wrong. Epsilon is initialized with the empty set on line 37, but never seems to become something different than the empty set. For every t, we iterate over elements of epsilon on line 41. However since epsilon is empty this is skipped. Then, in line 46 we assign the value of epsilon2 to epsilon, which is empty again, and we continue. So this case always returns an empty set. Typos: - line 28: "a maximum and minimum limit" or "maximum and minimum limits" - 69: "...a more flexible way..." - 76: "..physical world, or rules followed..." - 105: ".. and are not necessarily the same..." - 132: subscripts for omega should be superscripts. - 154: subscripts for omega should be superscripts. - 178: subsection - 194: "that are connected" - 256: dropping


Review 3

Summary and Contributions: The paper introduces STLnet. A student-teacher based approach where i) a teacher network generates traces satisfying the Signal Temporal Logic (STL) specification that are closest to a student-networks predictions, ii) these traces are then used to guide the student network to learn to satisfy the specifications better. The approach is then demonstrated on two tasks, with promising evidence regarding the utility of incorporating specification based side-information into the training procedure.

Strengths: *The paper is technically correct, and the propositions are valid. The paper has interesting components, and considers a diverse set of specifications. *The paper is clear, well-written and combines two diverse domains (specifications inspired by the CPS community + deep learning).

Weaknesses: *Temporal logic as such is often useful when considering infinite traces. Signal temporal logic is quite useful in the case of finite-time traces when dealing with continuous time systems. Neither of them are under consideration here, and I think this is the biggest draw back. *Novelty is quite limited. Much of the ideas in the paper have been introduced before. I will list some out here (which hasn't been discussed in the paper): 1. Writing STL specifications in terms of DNF specifications/using logical operators has been done previous in works such as: a) https://arxiv.org/abs/1703.09563 (they use DNFs/MILPS, which are equivalent), b) https://openreview.net/forum?id=BklC2RNKDS *Projecting outputs to satisfy constraints has been considered in works such as: a)https://arxiv.org/pdf/1805.07075.pdf, b) https://arxiv.org/abs/1801.08757 c)https://arxiv.org/abs/1603.06318. In fact, if you're only looking at finite+discrete time traces, this work is quite similar to c. Without considering infinite traces, the STL specifications reduce to first-order logic specifications as considered in c already. 2. There is a rich history of consider temporal logic specifications, and even STL specifications in the CPS community. As such, the specifications introduced here are not novel/new. 3. Propositions 4.1, 4.2 and 4.3 are trivial/obvious to the best of my knowledge. 4. There's very little discussion in terms of related work regarding enforcing STL/TL specifications for learning systems -- there have been several papers attempting to do this. a) https://robotics.sciencemag.org/content/4/37/eaay6276.full b) https://dorsa.fyi/publications/sadigh2014learning.pdf c) https://dl.acm.org/doi/abs/10.5555/3306127.3331994 d) https://openreview.net/forum?id=BklC2RNKDS

Correctness: The propositions are correct, and the experiments are thorough.

Clarity: Yes, the paper is quite clear. Typo: Line 171, what is Step 4 here?

Relation to Prior Work: I think a whole spectrum of related work has not been discussed in this work. I have included several pointers in an earlier section. I list some pieces of related work that are very closely related but haven't been included: a) https://arxiv.org/abs/1603.06318 (Projects to a prediction satisfying a constraint, which is then used to teach a student network -- has a very similar teacher/student framework). b) https://openreview.net/forum?id=BklC2RNKDS (enforced STL specifications for RNNs/LSTMs) b) https://www.sciencedirect.com/science/article/pii/S240589631831139X (enforces STL specifications for DNN controllers)

Reproducibility: Yes

Additional Feedback: I have read the rebuttal and would like to stay with my current score as most of my concerns have not been addressed.


Review 4

Summary and Contributions: This paper is on the problem of integrating temporal logic knowledge/constraints with the training of recurrent networks (RNNs) to encourage the RNNs to produce logically meaningful predictions. The framework first formulates the temporal knowledge using signal temporal logic (STL). Then, during the RNN training, a teacher model is constructed which produces a sequence prediction that is closest to the target (student) model's prediction while satisfying the given STL. The teacher's prediction is then used as supervision for updating the target model. Experiments on synthetic data and a real air quality dataset show the proposed approach achieves stronger and more meaningful results than the vanilla RNNs.

Strengths: - The problem of integrating structured/logical knowledge with deep NNs is of great practical signfiance yet under-explored - The proposed approach that uses STL for temporal knowledge encoding, and constructs teacher models for supervising the target student model, looks sound - Expriments show improvement over the vanilla LSTM on the real air quality dataset

Weaknesses: - The framework has essentially the same structure with the one developed in [7], which also extends knowledge distillation [6] by first constructing a teacher model that integrates the target student's prediction with the logical knowledge, then distilling the info from the teacher to the student. The differences are: * This paper considers temporal logic knowledge encoded with STL, instead of the first-order logic considered in [7] * The construction of teacher model is to an extent different. This paper is by manipulating the student's prediction according to the given STL, while [7] uses posterior regularization. The technical contribution is thus less impressive. Such a discussion about [7] is missing in the paper. - The experiments are a bit weak, since only one real dataset is used, and the comparion is only with the vanilla LSTM model. How about other stronger baselines, such as attentional LSTM which is very popular with lots of open-source packages? It's interesting to see how/whether the proposed approach could be applied and improve over the slightly more advanced NN architectures. - The STL trace generation in section 4.2 involves quite a few steps. What's the computation complexity of the whole process?

Correctness: Correct

Clarity: Yes

Relation to Prior Work: More discussions about [7] is necessary, given the similarity of the algorithmic framework

Reproducibility: Yes

Additional Feedback:

[Author Response · NeurIPS 2020]

We appreciate the valuable comments from the reviewers. We will answer reviewers' questions from three aspects, i.e.,
the novelty of the paper, algorithm scalability, and model properties. Due to the page limit, we will also address the
other comments in the paper if accepted.

**Novelty:** In respond to *Reviewer 5*, this paper's major novelty is developing a new STL-based learning framework to
enforce multivariate RNN models to follow critical model properties, especially targeting the sequential regression
tasks. Our method creates a practical way to ensure the logic rules' satisfaction in an end-to-end manner. It increases
the robustness of the RNN models. Our approach achieves promising results on real city datasets, i.e., significantly
increasing the satisfaction of model properties (by about four times) and prediction accuracy (by about 18.5%).

We have carefully compared our work with all the related papers pointed out by the reviewers. *First,* STL, as a powerful
specification language, has been broadly applied to the specification and verification for CPS applications, such as
robotics [1,2], smart cities, healthcare. Therefore, we also choose STL to express the model properties. STL has been
applied to both continuous and discrete signals. Due to the nature of RNN, the traces in this work are discrete with a
finite length. Using STL to specify CPS properties is not our novelty. However, we systematically identify six critical
types of model properties in CPS, which we believe is valuable for users to define model properties in their context
and utilize our work in practice. *Second,* DNF and many equivalent forms have been used in different contexts. Our
algorithm not only converts STL to DNF, also calculates the satisfaction range for each predicate and thus finds the best
trace closest to a given trace. Besides, we also create algorithms to generate satisfaction traces tailored to deep learning
processes efficiently. *Third,* introducing formal logic to support learning has been a hot topic and achieved promising
performance in recent years, including our work. Most of the current works focus on reinforcement learning [3,4] and
classification tasks [5,6], which have very different scopes than our paper. Their proposed methods do not apply to our
target problem. For example, paper [5] (already cited in our paper) combines first-order logic with neural networks
using a Teacher-Student network structure targeting NLP (classification) tasks. Paper [7] (a paper rejected by ICLR
2020) does apply to RNN models. It adds a term of constraint to the loss function, and tries to reach globally minimal
robustness over the input space. However, it is much more time-consuming and less robust (a soft constraint enforced
by optimization) comparing to our teacher-student network structure. Different from these papers, our work targets
multivariate RNN-based regression tasks, uses a more representative logic for RNN training, and achieves a stronger
satisfaction of the requirement (satisfaction guarantee with the teacher network at the testing time).

**Scalability**: In respond to *Reviewer 1*, the computation time of Algorithm 1 is relevant to the number of predicates in
the STL formula. However, we create algorithms to generate satisfaction traces tailored to deep learning processes
efficiently. The time could increase when there are more predicates, but Algorithm 1 only needs to be executed ONCE
in the pre-process (i.e., before the training phase). Therefore, it will not cause any significant delay in training and
testing phases, even for a large amount of data or long-term prediction. Besides, there are approaches to obtaining
a sub-optimal solution in a reasonable time that can be integrated to Algorithm 1 if needed. In our evaluation, the
pre-processing time for all cases (which have reasonable complexity STL formulas as the real-world applications) is
less than 10 seconds. We will also address it in the paper.

To briefly answer the other questions from Reviewer 1, (1) the reviewer is right about the teacher network; (2) The
return value is a non-negative real number. If a variable satisfies a constrain in a clause, the term will be evaluated to
0; Otherwise, it will return the minimal distance over all the items in the satisfaction of $l_i$ (not necessary to be 1). (3)
STLnet is general enough to be applied to transformer-based sequence models. Choosing RNN and its variants is to
show the generalizability of our solution.

**Model properties:** In respond to *Reviewer 4*, Model properties broadly exist in real-world applications and systems. In
this paper, we identify several critical types (in Section 2 and evaluation) based on the model properties from existing
papers, systems, and applications in CPS domain. In practice, model properties can be (1) already known by the system
before prediction, e.g., constraints by the physical world, rules followed by the application domains, (2) defined by the
users based on their application (e.g., robotics), (3) mined from the models' historical behaviors. (We also present a
similar discussion at the beginning of Section 2 in the paper.)

To briefly answer the other questions from Reviewer 4, (1) RMSE itself cannot capture the temporal correlations of the
sequence like eventually, existence, consecutive changes, etc. (2) [0,24] represents 24 hours in a day. Users can choose
to use () or [] based on if the beginning and ending hours are included. (3) Alg. 1 has a typo that the epsilon set should
be initialized with CalculateDNF($\phi_1, t, sgn$) where $t$ is an element of $T$.

**References**:
[1] Raman, Vasumathi, et al. "Model predictive control with signal temporal logic specifications." 53rd IEEE Conference on Decision and Control. IEEE, 2014.
[2] Dutta, Souradeep, et al. "Learning and verification of feedback control systems using feedforward neural networks." IFAC-PapersOnLine 51.16 (2018): 151-156.
[3] Li, Xiao, et al. "A formal methods approach to interpretable reinforcement learning for robotic planning." Science Robotics 4.37 (2019).
[4] Sadigh, Dorsa, et al. "A learning based approach to control synthesis of markov decision processes for linear temporal logic specifications." 53rd IEEE CDC. 2014.
[5] Hu, Zhiting, et al. "Harnessing deep neural networks with logic rules." arXiv preprint arXiv:1603.06318 (2016).
[6] Ghosh, Shalini, et al. "Trusted Neural Networks for Safety-Constrained Autonomous Control." arXiv preprint arXiv:1805.07075 (2018).
[7] Dathathri, Sumanth, et al. "Scalable Neural Learning for Verifiable Consistency with Temporal Specifications." (2019).


[Meta-Review · NeurIPS 2020]

This paper initially received three reviews. The reviewers appreciated the integration of temporal logic with deep learning presented in the paper. The main concerns centered around the relations of the proposed method to existing literature and first-order logic specifications. After reading the authors' rebuttal the reviewers engaged in detailed debate around the merits of the paper. A fourth expert reviewer's opinion was sought to help come to a decision. In the end, it was determined that the merits of the paper outweigh the potential concerns. This is an interesting, important area of recent research and the paper will continue to spur interest in this direction. The authors are encouraged to modify the camera ready to: include further discussion regarding FOL vs. the proposed STL representation, the relation to previous work cited by the reviewers and particularly a thorough discussion with respect to [7].